# Thermal, Mechanical, and Microstructural Study of PBO Fiber during Carbonization

**DOI:** 10.3390/ma12040608

**Published:** 2019-02-18

**Authors:** Weizhe Hao, Xuejun Zhang, Yanhong Tian

**Affiliations:** 1State Key Laboratory of Organic-Inorganic Composites, Beijing University of Chemical Technology, Beijing 100029, China; 2016200566@mail.buct.edu.cn (W.H.); tianyh@mail.buct.edu.cn (Y.T.); 2Key Laboratory of Carbon Fiber and Functional Polymers, Ministry of Education, Beijing University of Chemical Technology, Beijing 100029, China

**Keywords:** PBO, carbon fiber, carbonization, continuous process

## Abstract

Poly(p-phenylene benzobisoxazole) (PBO) fiber shows fascinating properties including excellent mechanical performance, high crystallinity, and fairly good heat resistance as a kind of polymer fiber. Its properties make it a possible candidate as a precursor of carbon fiber. This paper mainly investigates the possibility of yielding carbon fiber from PBO by direct carbonization using a continuous process and multiple properties of yielded fiber treated under different heat treatment temperature (HTT). The results show that PBO fiber was able to sustain an HTT as high as 1400 °C under the inert atmosphere and that the shape of fiber was still preserved without failure. Using thermal gravimetric analysis (TGA) and TGA coupled with mass spectroscopy (TGA-MS), it was found that a significant mass loss procedure happened around 723.3 °C, along with the emission of various small molecules. The mechanical performance first suffered a decrease due to the rupture of the PBO structure and then slightly increased because of the generating of graphite crystallite based on the broken structure of PBO. It was observed that PBO’s microstructure transformed gradually to that of carbonaceous material, which could be the reason why the change of mechanical performance happened.

## 1. Introduction

Carbon fiber has been paid much more attention recently due to its excellent properties such as high tensile strength, tunable Young’s modulus, extraordinary chemical stability, and heat resistance [1,2,3,4], which has broadened its application field to include aerospace, commercial industries, etc. [5]. This material is suitable as a kind of fiber reinforcement for producing fiber reinforced plastics (FRP) especially [6,7,8].

Currently, carbon fiber is yielded substantially by heat treatment of polymer precursor [7,9]. The commonly used precursor contains polyacrylonitrile (PAN) [10], pitch [11,12,13], and rayon [7,14,15]. Among those polymer precursors, PAN is mostly adopted to produce carbon fiber, and PAN-based carbon fiber (PAN-CF) takes up approximately 90% of the global market [10]. Traditional PAN-CF techniques mainly contains three steps: pre-oxidation, carbonization, and graphitization [16]. During the pre-oxidation process, cyclization reaction happens in the chain structure of PAN to form a ladder structure, which ensures the fiber to sustain higher temperatures during the carbonization and graphitization processes [17,18]. The manufacturing of a PAN precursor is a high-cost step which takes up over 50% of the production cost of PAN-based carbon fiber [19]. Also, the pre-oxidation process requires prolonged treating time and massive energy, which makes this procedure take up approximately 20% of the total cost [20].

Since lowering costs is essential for carbon fiber industries, solutions are needed to lower costs. One approach to solving this problem is to find a new type of polymer precursor. Poly(p-phenylene benzobisoxazole) (PBO) fiber seems to be a possible candidate. PBO fiber has excellent mechanical properties, high crystallinity, and fairly good heat resistance [21,22], which means it substantially meets the requirements of being a precursor of carbon fiber. However, PBO suffers a great deal from aging brought on by specific conditions, such as prolonged exposure to UV light. It has been reported that the tensile strength of PBO fiber can dramatically decrease due to the UV-aging process [23]. So far, researchers have studied the heat treatment of PBO fiber for the purpose of improving its mechanical properties [24,25] or modifying its surface as a reinforcing phase in composite material, but limited research has been conducted towards PBO-based carbon fiber (PBO-CF) [26,27,28], especially with aged PBO fiber as the precursor. For this reason, the possibility of yielding carbon fiber from a PBO precursor needs to be studied.

Using aged PBO fibers as the raw material, the purpose of this paper was to study the mechanism of PBO during carbonization process as well as the capability of obtaining carbon fiber through heat treatment of a PBO fiber.

## 2. Materials and Methods

Aged PBO fibers were first cut up into pieces with lengths of up to 2 mm for thermal analysis. A NETZSCH STA 449F3 analyzer (NETZSCH-Gerätebau GmbH, Selb, Germany) was used to conduct thermal gravimetric analysis (TGA) under a nitrogen atmosphere. The temperature range was set as 40–1300 °C, with a heating rate of 10 °C·min^−1^ and nitrogen gas flow of 100 mL·min^−1^. A PerkinElmer Pyris Diamond TG/DTA analyzer (PerkinElmer Inc., Waltham, MA, USA) and an OmniStar^TM^ mass spectrometer (Pfeiffer Vacuum GmbH, Asslar, Germany) were used in TGA coupled with mass spectroscopy (TGA-MS) under an argon atmosphere. The temperature range was set as 30–1150 °C, with a heating rate of 10 °C·min^−1^ and an argon flow rate of 100 mL·min^−1^. Possible emission particles with relative molecular mass less than 200 Da in the outflow were detected by the mass spectrometer.

During the direct carbonization process, PBO fiber was treated by a self-built production line in a continuous rather than batch manner. The soaking time was set to 2 min, while the heat treatment temperature (HTT) was set to 700, 750, 800, 850, 900, 1100, 1300, and 1400 °C, respectively. Samples were collected for further investigation.

Mechanical properties of each sample including tensile strength and Young’s modulus were measured using an INSTRON 3345 universal testing system (INSTRON, Norwood, CO, USA).

Organic elemental analysis of those yielded samples was performed on a varioEL III organic element analyzer in CHN mode. The relative content of each element was investigated.

The diameter of each sample was measured using an optical microscope. Scanning electron microscopy was conducted on a JEOL JSM-6701F scanning electron microscope (JEOL Ltd., Akishima, Tokyo, Japan).

An XRD analyzer (BRUKER D8 ADVANCE) (Bruker Corporation, Billerica, MA, USA) was used to conduct analyses of those samples. The obtained XRD curves were processed using Jade 6.5 software (Materials Data Inc., Livermore, CA, USA) to obtain information about the crystalline structure of those samples.

Raman spectroscopy was performed using a Renishaw inVia spectrometer (Renishaw plc, Wotton-under-Edge, UK). The laser wavelength was 514.5 nm and the detecting range was set as 800–2000 cm^−1^. The obtained spectra of those samples were imported in Origin 2018 software (OriginLab Corporation, Northampton, USA) for further investigation.

## 3. Results and Discussion

### 3.1. Thermal Analysis of PBO Fiber

TGA was firstly conducted on cut-up PBO fiber to investigate its thermal behavior during the carbonization process. As is shown in Figure 1, TG and the derivative of the TG (DTG) curves were obtained for PBO fibers heated up to 1300 °C under an inert atmosphere. The TG curve reflects the mass loss of the sample directly and the DTG curve is used to determine the mass loss peak. In this case, two peaks at 168.5 °C and 723.3 °C can be observed, which indicate that there were two main mass loss procedures during the whole process. Of note is that the second mass loss was much more rapid and more severe than the first one, which indicates that the pyrolysis reaction during the second mass loss was more violent than that during the first mass loss. Furthermore, the mass loss did not stop but went on at a fairly slow rate as the HTT went above 850 °C until the analysis terminated. One can infer that the mechanics of the pyrolysis reaction above 850 °C were different from those during the second mass loss.

Because the reaction during the second mass loss was the more complex one, further investigation had to be conducted. Thermal gravimetric analysis coupled with mass spectroscopy was used to examine the existence of several kinds of possible molecules evacuating from PBO fiber during the carbonization process. Due to the limitations of the equipment, the target temperature was set to 1150 °C. The results are shown in Figure 2. It should be pointed out that the peaks existing in those graphs suggest the presence of certain kinds of particles in the outflow under certain temperatures. Here, the positions of all of the peaks were located in the range of 750–760 °C. It could be inferred that these peaks corresponded to the main mass loss of PBO with a minor and acceptable shift in the case of temperature.

Furthermore, the mechanism of the possible reaction during the carbonization of PBO fiber can be discussed regarding the structure of PBO’s polymer chain. Figure 3 shows a monomer which is part of the structure of the PBO chain, and some of the chemical bonds are labeled from 1 to 6. As is mentioned above, the presence of HCN, NO, CO_2_, benzene, and benzonitrile were confirmed in the outflow. Firstly, because benzene shows much better aromaticity than oxazole, the carbon atoms on the two benzene rings are steadier than those on the two oxazole rings. Therefore, it could be predicted that the origin of the carbon atoms in those small molecules is the oxazole rings rather than the benzene rings in the PBO chains. The homolytic scission of bond 1 and 4 would lead to the generating of a nitrile bond (C≡N), and the breaking of bond 5 could be one of the conditions of the emission of HCN. The generating of CO_2_ requires the scission of bonds 2, 3, and 5, which would provide one carbon atom and one oxygen atom. In addition, it also needs one oxygen atom from the adjacent PBO chain. As for the generation of NO, it is certain that the scission of bonds 1 and 3 can provide N atoms and the scission of bonds 2 and 4 can provide O atoms. However, the N atom and O atom from those NO molecules seem not to originate from the same oxazole ring. If so, two radicals in the benzene ring and one carbyne-like radical would be generated with the scission of bonds 1 to 4, which requires massive energy and seems impossible. Therefore, it is believed that the NO molecules originate from the multi oxazoles. Finally, the origination of benzene and benzonitrile is more obvious than for that of the other molecules. After the break of bond 5 and 6, there still needed to be two hydrogen atoms from the adjacent PBO chain to form a benzene molecule. As for benzonitrile, bonds 1, 4, and 6 needed to be broken to generate its backbone. 

Also, the molecules H_2_, C, NH_3_, and C_2_H_2_ were not expected in the outflow, but their existence can be discussed here. With the scission of bonds 3, 4, and 5, a single C atom may release from the chain structure along with the chain break. Considering the fact that PBO has a deficiency of hydrogen, H atoms in H_2_, NH_3_, and C_2_H_2_ molecules are not likely to originate from the same chain but may be generated from the interchain reaction. Otherwise, exceeding radicals would be formed in a monomer part of the PBO chain.

The emission of all kinds of small molecules surely leads to the rupture of the chain structure of PBO. However, some kinds of small molecules are surely generated from the interchain reaction, and the radicals generated due to the reaction may trigger the interchain reaction, which prevents further rupture of the fiber and could be the reason of the mass remaining being more than 50%. One could also infer that the emission of those molecules also creates the conditions for the formation of a graphite crystallite structure, which will be discussed later.

### 3.2. Mechanical Performance of PBO-Based Carbon Fiber

Since PBO fiber can sustain high temperatures during the carbonization process, samples of PBO-based carbon fiber were collected by a continuous method. The mechanical performance of each sample was then tested using an INSTRON 3345 universal testing system. The results are shown in Table 1. It is demonstrated that tensile strength and Young’s modulus were affected greatly by HTT. Tensile strength reaches its minimum of 350 MPa at 750 °C. After that, tensile strength starts to increase until the HTT reaches 1100 °C. From that point, the increase in tensile strength is negligible. The situation is similar when it comes to Young’s modulus of the sample. Young’s modulus also reaches its minimum of 53 GPa at 750 °C and then increases to 130 GPa at 1100 °C. The severe decrease in mechanical performance implies that the pyrolysis reaction was vigorous at 750 °C, during which the rupture of the microstructure of PBO fiber took place, along with the emission of various molecules. Moreover, the slight increase in mechanical performance at higher temperatures may also prove that structural change occurred above 750 °C.

The PBO fiber could have undergone the carbonization process, with mechanical performance corresponding to that of general-purpose carbon fiber. Still, this was not ideal, since tensile strength and Young’s modulus of PAN-based carbon fiber can reach 2 GPa and 200 GPa, respectively. This kind of situation resulted from the change in microstructure of the material during heat treatment, which still needs further discussion. If the ladder structure in pre-oxidized PAN fiber is considered a chain, the carbonization of PAN fiber is a kind of interchain reaction. There is no doubt that this reaction leads to a large graphite-like structure, which is the main reaction during the carbonization process. When it comes to PBO, however, the reaction is quite complicated, with both intrachain and interchain reactions happening during carbonization. It is believed that these complicated reactions result in a ruptured carbonaceous structure unlike with what happens to PAN fiber, and this would be the reason why the mechanical performance of PBO-derived carbon fiber only meets the requirement of general-purpose carbon fiber.

### 3.3. Evolution of Elemental Composition of PBO Fiber during the Carbonization Process

To investigate the evolution of elemental composition during the carbonization process, all samples were characterized using organic element analysis (OEA), and the ratio of carbon content to hydrogen content (C/H), nitrogen content (C/N), and oxygen content (C/O), respectively, was calculated. The results are shown in Table 2. The element composition seemed to change smoothly, with carbon content increasing and other non-carbon element content decreasing while the HTT went up. When HTT reached 1400 °C, the carbon content reached its maximum of 95.76 wt.%. The situation is similar when it comes to C/H, C/N, and C/O. The values of all mentioned ratios reached their maxima at 1400 °C, and were 229.1, 40.21, and 45.82, respectively. It can be inferred that during the carbonization process of PBO fiber, non-carbon elements including hydrogen, nitrogen, and oxygen tended to evacuate the bulk in the form of several kinds of molecules.

### 3.4. Change of Microstructure of PBO Fiber during Carbonization Process

Diameters of all samples were measured and the results are shown in Table 3. They indicate radial shrinkage during heat treatment. First, a decrease can be observed with increasing HTT. At 1400 °C, the diameter had reduced by 21.62% compared to that of the untreated fiber. Moreover, in the temperature range 750–1100 °C, the decrease was the most significant, which was consistent with the result of the mechanical study. The evacuation of the non-carbon elements could be the reason why PBO fiber shrank during heat treatment.

Scanning electron microscopy (SEM) was conducted on those samples and the images are shown in Figure 4. The brightness of each image indicates the electrical conductivity of raw or treated fiber. It can be seen that fibers which were treated under higher temperature are brighter than those treated under lower temperature. Of note is that the brightness of the SEM image is determined by the electrical conductivity of the sample. Considering that the conductive coating can affect the conductivity as well as the morphology, the samples were not sprayed, leaving the conductivity unchanged. From these SEM images, increasing electrical conductivity can be observed with increasing temperature. Since the graphite-like structure can improve the conductivity of the material, it can be inferred that this kind of structure formed and developed due to high-temperature treatment, and was more developed when the HTT increased.

To study the change in the crystalline structure of PBO fiber during the carbonization process, XRD analysis was conducted on each sample respectively, and XRD patterns are shown in Figure 5. Patterns of untreated, 700 °C-, and 750 °C-treated fibers show a typical (200) peak of PBO crystallite [29], while those of the 800 °C-, 850 °C-, 900 °C-, 1100 °C-, 1300 °C-, and 1400 °C-treated fibers show a typical (002) peak of graphite crystallite [30]. This could be evidence of structural transformation taking place in PBO fiber during the carbonization process.

Further information was acquired using Jade 6.5 software (Materials Data Inc., Livermore, CA, USA). Peak positions and full width at half maxima (FWHM) were obtained by a peak fitting process. Moreover, the interplanar distance and crystallite size were calculated by the Bragg equation and Scherrer formula.
(1)$$d=\frac{\lambda }{\sin\theta }$$
(2)$$L=\frac{K\lambda }{\beta \cos\theta }$$

In these equations, *d* (*d*_200_ or *d*_002_) is the interplanar distance of PBO or graphite crystallite; *λ* is the wavelength of Cu K*_α_*_1_ (*λ* = 0.15406 nm); *θ* is the diffraction angle; *L* (*L*_⊥200_ or *L*_⊥002_) is the size in the direction perpendicular to the crystal face; *K* is Scherrer constant (*K* = 0.89); and *β* is the FWHM of the diffraction peak.

All of the results are shown in Table 4. It is obvious that *L*_⊥200_ increased from 5.263 nm to 8.924 nm with *d*_200_ remaining almost unchanged when the HTT reached 700 °C. This could be evidence of heat-induced recrystallization which could increase Young’s modulus of the PBO fiber. However, *L*_⊥200_ decreased dramatically with further heat treatment, and the (200) peak even disappeared when HTT reached 800 °C. This is evidence proving that the rupture happened in the PBO crystallite during heat treatment. The appearance of a (002) peak of graphite from 800 °C indicated that graphite crystallite was generated based on the broken structure of PBO crystallite. With a further increase in HTT from 800 °C, *d*_002_ dropped slightly from 0.355 nm to 0.348 nm while *L*_⊥002_ increased from 1.132 nm to 1.552 nm. The result still proves that heat treatment under higher HTT improves the structure of graphite crystallite.

Since the transformation of crystallite from PBO to graphite took place in the treated fiber, Raman spectroscopy was applied to further study the change in the degree of graphitization of those samples. Spectra of those samples except for the one belonging to the 650 °C-treated fiber are shown in Figure 6. The spectrum of the 650 °C-treated fiber was not obtained because of the fluorescence effect during the analysis process. Still, the characteristic peaks of PBO at 930, 1170, 1290, 1305, 1540, and 1620 cm^−1^ can be observed clearly in the spectra of untreated and 700 °C-treated fiber [31]. From the spectrum of the 750 °C-treated fiber, it could be inferred that the transformation of the structure of crystallite in fiber was happening due to heat treatment. The rest of the spectra showed typical graphite peaks of a D peak (located around 1370 cm^−1^) and G peak (located around 1595 cm^−1^) [32], which prove the appearance of the graphite structure.

More information was acquired from data analysis using Origin 2018 software. By a multi-peak fitting method, peak position and peak area were obtained. Furthermore, the area ratios of the D peak to G peak (*R*) of 800 °C-, 850 °C-, 900 °C-, 1100 °C-, 1300 °C-, and 1400 °C-treated samples were calculated. The results are shown in Table 5. The positions of both the D and G peaks almost stayed unchanged while the HTT increased, while *R*’s value decreased slightly from 3.423 at 800 °C to 2.467 at 1400 °C. Since a lower *R* suggests a higher degree of graphitization, the situation mentioned above suggests that a higher HTT led to a higher degree of graphitization during the carbonization process of the PBO fiber.

## 4. Conclusions

In this paper, the possibility and mechanics of yielding carbon fiber from a PBO fiber precursor by direct carbonization using a continuous process was investigated. PBO fiber was able to sustain heat treatment under a nitrogen atmosphere up to 1400 °C. During the carbonization process, PBO crystallite transformed into graphite crystallite spontaneously and gradually, in addition to various small molecules evacuating the treated fiber. The mechanical performance met the needs of general-purpose carbon fiber, and thus the possibility of using aged PBO as a kind of precursor for producing carbon fiber was confirmed.

## Figures and Tables

**Figure 1 materials-12-00608-f001:**
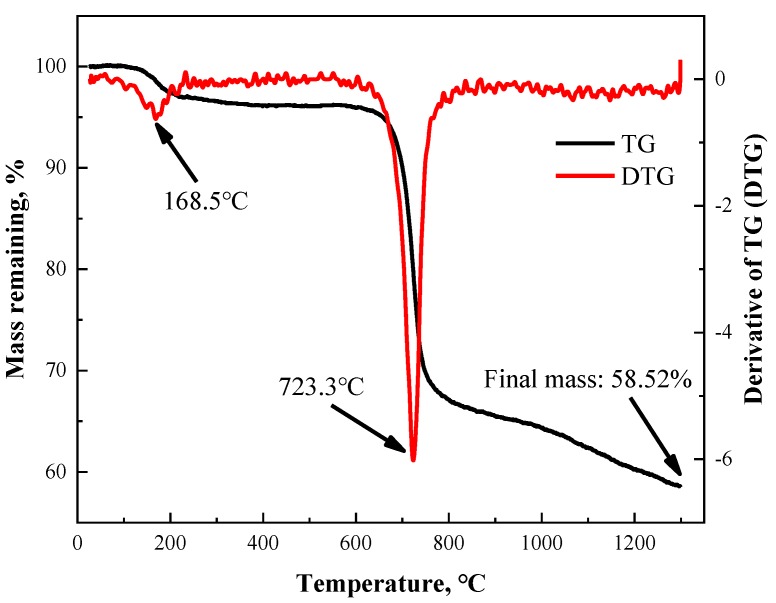
Thermalgravimetric analysis (TGA) results of poly(p-phenylene benzobisoxazole) PBO fiber heated up to 1300 °C under an inert atmosphere.

**Figure 2 materials-12-00608-f002:**
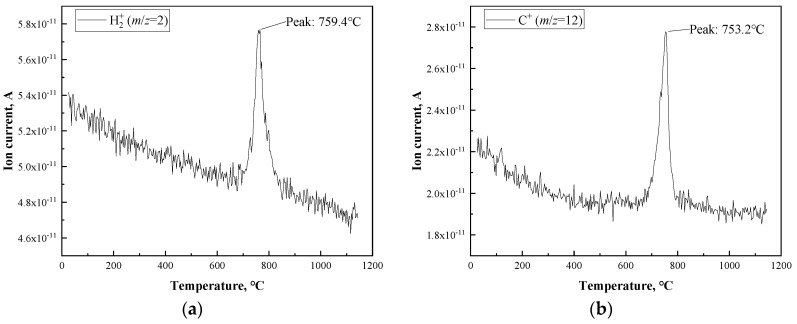
Thermalgravimetric analysis coupled with mass spectroscopy (TGA-MS) results of PBO fiber heated up to 1150 °C under an inert argon atmosphere. (**a**) H${}_{2}^{+}$; (**b**) C^+^; (**c**) CH${}_{4}^{+}$; (**d**) NH${}_{3}^{+}$; (**e**) H_2_O^+^; (**f**) C_2_H${}_{2}^{+}$; (**g**) HCN^+^; (**h**) CO^+^ or N${}_{2}^{+}$; (**i**) NO^+^; (**j**) O${}_{2}^{+}$; (**k**) CO${}_{2}^{+}$; (**l**) NO${}_{2}^{+}$; (**m**) benzene ion; (**n**) benzonitrile ion.

**Figure 3 materials-12-00608-f003:**
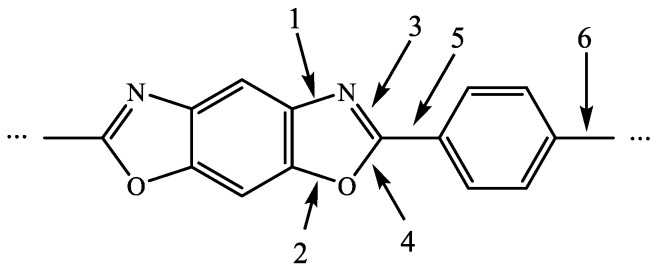
A monomer part of the PBO polymer chain.

**Figure 4 materials-12-00608-f004:**
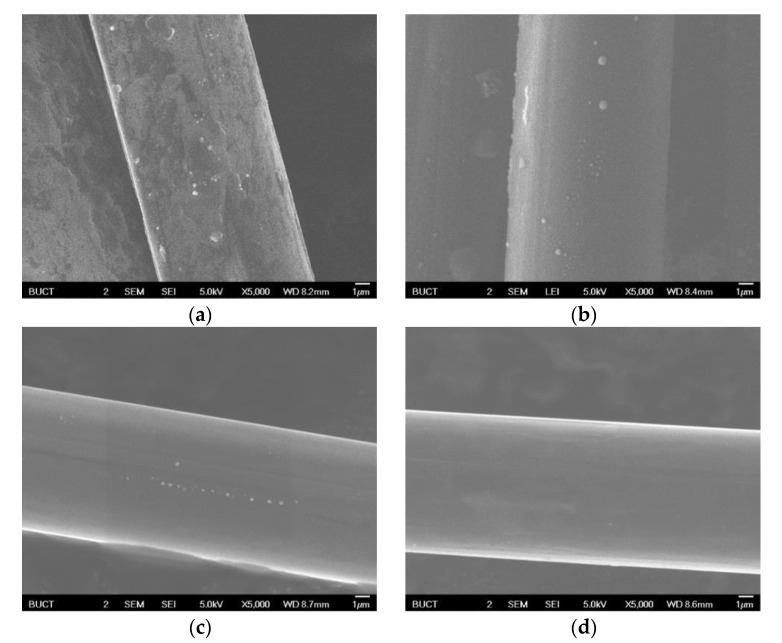
SEM images of each sample. (**a**) Untreated; (**b**) 700 °C-treated; (**c**) 900 °C-treated; (**d**) 1100 °C-treated.

**Figure 5 materials-12-00608-f005:**
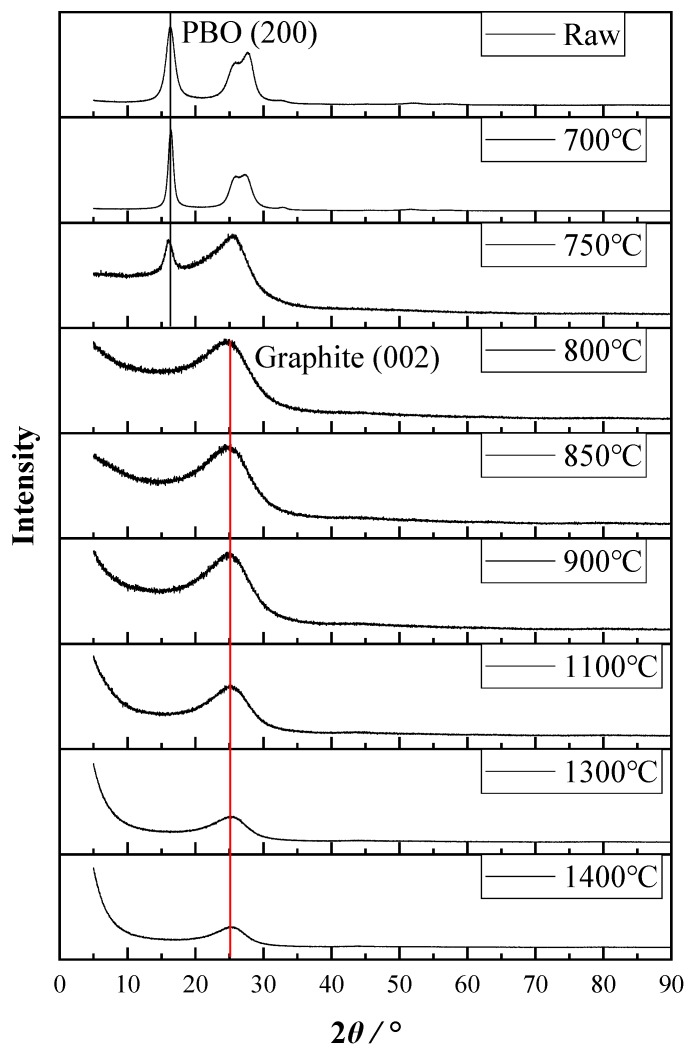
XRD pattern of each sample.

**Figure 6 materials-12-00608-f006:**
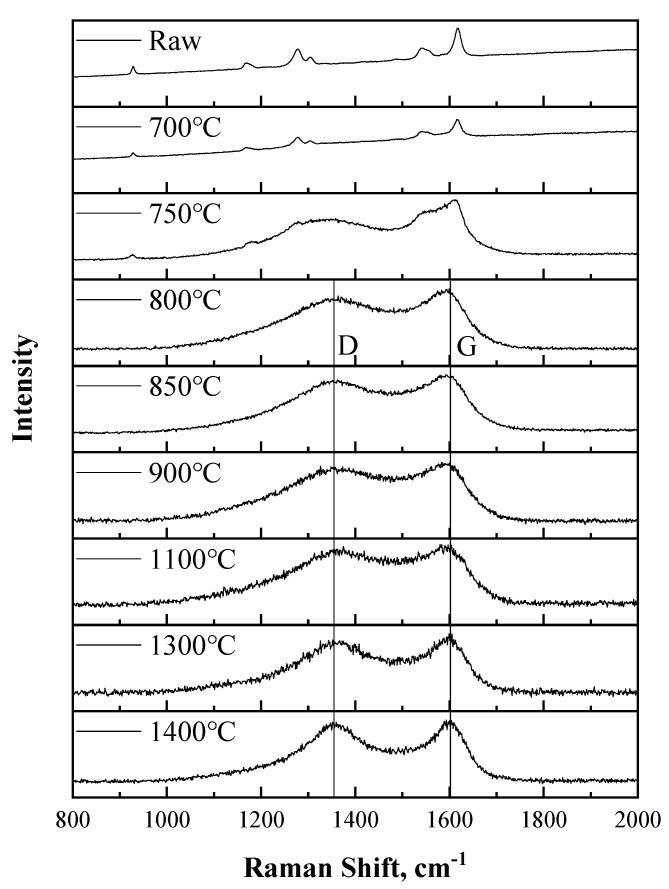
Raman spectra of each sample.

**Table 1 materials-12-00608-t001:** Mechanical performance of each sample. Legend: HTT, heat treatment temperature.

HTT (°C)	Tensile Strength (MPa)	Young’s Modulus (GPa)	Strain at Failure (%)
700	484 ± 41	116 ± 8	0.8
750	350 ± 38	53 ± 5	1.3
800	429 ± 35	65 ± 6	1.3
850	438 ± 33	75 ± 6	1.1
900	470 ± 41	92 ± 8	1.1
1100	516 ± 42	130 ± 8	0.9
1300	521 ± 40	131 ± 11	0.8
1400	523 ± 36	131 ± 10	0.7

**Table 2 materials-12-00608-t002:** Organic element analysis (OEA) results of carbonized samples.

HTT (°C)	Element Content (wt.%)				C/H	C/N	C/O
	C	H	N	O			
untreated	69.25	3.069	11.52	17.95	22.56	6.011	3.858
700	70.80	2.124	11.58	15.75	33.33	6.113	4.496
750	73.83	2.697	9.849	13.64	27.37	7.496	5.412
800	76.02	2.335	8.396	13.48	32.56	9.054	5.639
850	76.73	2.039	8.153	13.72	37.63	9.411	5.592
900	77.58	1.852	7.988	12.23	41.89	9.712	6.343
1100	83.39	1.163	6.554	8.548	71.70	12.72	9.755
1300	93.20	0.522	3.228	3.258	178.5	28.87	28.61
1400	95.76	0.418	2.381	2.090	229.1	40.21	45.82

**Table 3 materials-12-00608-t003:** Results of diameter measurement.

HTT (°C)	Diameter (μm)	Relative Reduction (%)
untreated	13.97	-
700	13.03	6.729
750	12.92	7.516
800	12.28	12.10
850	11.87	15.03
900	11.67	16.46
1100	11.05	20.90
1300	10.98	21.40
1400	10.95	21.62

**Table 4 materials-12-00608-t004:** XRD parameters obtained from data analysis. Legend: FWHM, full width at half maximum.

HTT (°C)	2*θ* (°)		*d* (nm)	FWHM (°)	*L* (nm)
	PBO (200)	Graphite (002)			
untreated	16.197	-	0.547	1.509	5.263
700	16.303	-	0.544	0.890	8.924
750	15.856	-	0.559	1.381	5.748
800	-	25.080	0.355	7.117	1.132
850	-	25.374	0.351	6.937	1.162
900	-	25.533	0.349	6.748	1.195
1100	-	25.690	0.347	6.170	1.307
1300	-	25.540	0.349	5.398	1.493
1400	-	25.602	0.348	5.195	1.552

**Table 5 materials-12-00608-t005:** Results obtained from fitting analysis for the Raman spectra.

HTT (°C)	D Peak Position (cm^−1^)	G Peak Position (cm^−1^)	*R*’s Value
800	1373	1592	3.423
850	1373	1593	3.297
900	1379	1592	3.164
1100	1380	1595	3.009
1300	1372	1597	2.737
1400	1370	1596	2.467

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
