# Peer review of "Thermal, Mechanical, and Microstructural Study of PBO Fiber during Carbonization"

_materials, 2019, doi:10.3390/ma12040608_

Reviewer 1 Report

The author of the paper has presented, PBO as precursor material for CF, however, the paper lacks substantial study required to justify as the precursor material. Below are some of the points the author should work on so the paper could be considered for review. Materials is a highly accredited journal and this paper should be improved drastically.

Abstract is weak. Adjectives are not acceptable in the abstract. Line 13, "relatively high crystallinity", compared to what?

The microstructure of the fibers is missing. SEM, fiber cross section images need to be studied.

References need to be updated. There are carbon fiber review articles published in 2017 and above, need to cite more recent carbon fiber review articles than 18 to 20 years old.

How will the PBO fibers affect the cost of the carbon fibers? Percentage required and how it compares with the other low cost precursors such as textile grade PAN? There are references about textile grade PAN fiber precursors for CF synthesis.

Why are aged PBO fibers required? 

Line 64, what is 200? Is it Daltons?

Line 66, what is self-built?

Table 1. The increase in tensile strength, 523 MPa, is not significant after treating the fibers up to 1400 C. What is the significance of the material to be used as CF precursor when there are textile grade PAN precursor based CF which deliver up to 2.5 GPa

SEM imaging of the fibers are required to compare and assist the structural modification taking place at various temperature regions.

Is there any skin-core effect observed during carbonization process?

Author Response

Dear reviewer,

We would like to thank you for reviewing our manuscript and providing several constructive opinions. We have revised our paper, and attachment file is our point-to-point response.

Sincerely yours

Reviewer 2 Report

This paper addresses a new and cost-efficient method to fabricate carbon fibers from PBO fibers. It is true that a typical carbonization is rather an energy-intensive process, especially with the preoxidation (or stabilization) step. The authors have designed a more direct and seemingly energy-efficient process by simply carbonizing PBO. I believe this paper is a good fit to be published in Materials journal with some clarification and corrections.

1. Introduction

I agree with the authors that carbonization via PBO is a relatively direct and energy-efficient method to produce carbon fiber. However, my understanding is that fabrication of PBO itself is not a trivial process; its fabrication consists of cost-prohibitive polymerization and toxic solvent use. On the contrary, PAN fibers are not nearly as energy-intensive to make. It would be appropriate for the authors to at least mention this point, especially given the recent importance of sustainability of a process at all stages.

2. Materials and methods

What was the rationale behind the temperature range chosen for the carbonization? In other words, why 700- 1400 C? why not higher? A simple sentence or two should suffice.

3. Results and Discussion

I appreciate the author's rigorous thermal study which shed light on the carbonization mechanism of PBO at a molecular level. The data, in general, seem reliable. I do have a couple of comments for the mechanical properties.

a) Have the authors attempted to visually characterize these fibers? An SEM image, or at the very least, an optical microscope image of some sort, should shed light on i) the fiber morphology before and after carbonization, and ii) origin of failure/fracture after mechanical testing.

b) Mechanical properties of the carbonized fibers are not particularly impressive when compared to both PBO and carbon fiber; both of them typically exhibit strength and modulus well above 1.0-2.0 GPa and 200 GPa, respectively. At the very least, the authors should discuss/explain such deficiency.

Author Response

Dear reviewer,

We would like to thank you for reviewing our manuscript and providing several constructive opinions. We have revised our paper, and attachment file is our point-to-point response.

Sincerely yours

Round  2

Reviewer 1 Report

1.  There are grammatical errors and needs to fixed.

2.  Sentences should be more concise.

3.  In the introduction, reference 20 is not acceptable format. Here are some suggestions for the textile grade precursors for carbon fiber conversion and references about other precursors for carbon fiber synthesis: Also, the authors are in ORNL as the reference 20, so it will cover all the points.

a. Recent Developments in Carbon Fibers and Carbon Nanotube-Based Fibers: A Review (2017) Polymer Reviews

b. Polyacrylonitrile nanocomposite fibers from acrylonitrile-grafted carbon nanofibers (2017) Composites part B Engineering

c. Improving mechanical properties of carbon nanotube fibers through simultaneous solid-state cycloaddition and crosslinking (2017) Nanotechnology

4. Also, in the reference 19, cost of the carbon fiber is based on the precursor fibers not just stabilization process. Energy cost is still not as significant as the precursor fibers cost. Update the references with above suggested references which discuses about various cost aspects of carbon fiber synthesis and textile grade PAN precursor.

5.  Please add the SEM images in the paper and explain the surface morphology change with respect to the treatment temperature. How is the diameter of the fibers changing as the treatment temperature increases.

6. Add standard deviation to the mechanical properties of the fibers, without std deviation the data is not viable. Also, strain at failure is missing.

Author Response

Dear reviewer,

We have revised our manuscript based on your points. Thank you for your reviewing.

Sincerely yours
